# PlanTEA: Supporting Planning and Anticipation for Children with ASD Attending Medical Appointments

Patricia Hernández [1], Ana I. Molina [2,*], Carmen Lacave [2], Cristian Rusu [3] and Abel Toledano-González [4]

1    Asociación Regional de Afectados de Autismo y Otros Trastornos del Desarrollo (AUTRADE), 13004 Ciudad Real, Spain; patricia_autrade@hotmail.es
2    Escuela Superior de Informática, Universidad de Castilla-La Mancha, 13071 Ciudad Real, Spain; carmen.lacave@uclm.es
3    Escuela de Ingeniería Informática, Pontificia Universidad Católica de Valparaíso, Valparaíso 2340000, Chile; cristian.rusu@pucv.cl
4    Facultad de Ciencias de la Salud, Universidad de Castilla-La Mancha, 45600 Talavera de la Reina, Spain; abel.toledano@uclm.es
*    Correspondence: anaisabel.molina@uclm.es

**Abstract:** In people with Autism Spectrum Disorder (ASD), skills related to anticipation and mental flexibility are often impaired, so their thinking tends to be very rigid and their behavior is based on establishing routines. For this reason, children with ASD may show disruptive behaviors when faced with disturbing but necessary activities, such as going to a doctor's appointment. Therefore, it is very convenient and necessary for their families to prepare in advance for the visit and to explain the details of the procedure to be performed at the consultation. The use of anticipation boards in these situations allows to prepare such situations and to reduce stress for both the ASD child and their families or caregivers. In this context, the use of technology can provide great benefits for anticipating a new event, or whatever risks the control of their routines, as well as enhancing developmental skills such as communication, autonomy, social interaction, etc. This article describes a software tool, for mobile devices such as tablets, that allows the planning of the attendance of children with ASD to the necessary medical appointments throughout their childhood and adolescence, as well as communication with specialists. This app, named PlanTEA, has undergone a preliminary evaluation that has yielded very positive results. Most participants found the app useful in helping to anticipate (94.1%) and improve communication (94.2%) for people with ASD in medical contexts, considered it easy to use, with no technical support needed to use it (almost 100%), and would recommend its use (94.2%). This first evaluation has also allowed us to define the next steps to be taken to improve and enhance this tool and thus reach a wider population within the autistic disorder. As a result of the evaluation carried out and the comments received, it is proposed to extend its use to adult users and those with high-functioning autism, which supposes expanding and extending the functionalities of the current version of PlanTEA.

**Keywords:** Autism Spectrum Disorder (ASD); children; medical appointments; mobile application; usability; user experience (UX)

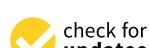



## 1. Introduction

Autism Spectrum Disorder (ASD) is defined as a neurodevelopmental disorder with a strong genetic basis, which manifests itself from an early age and is characterized by a delay or impairment in the acquisition of skills in a wide variety of domains including aspects such as the presence of stereotyped behaviors, rigid routines, repetitive and restricted interests, as well as persistent deficits in socialization, reciprocal interaction and communication [1]. These alterations imply difficulties in brain development and functioning, not always showing as structural lesions, being probable a deficiency in the maturation

of fibers that alter connectivity and therefore an adequate integration in neuronal communication. It is one of the most frequent emerging neurodevelopmental disorders in early childhood, although it is not exclusive to infantile and juvenile age, causing a significant deterioration in the life of the person. The diagnosis of this disorder has increased considerably in the last 40 years [2]. According to a recent report by the World Health Organization, it is estimated that 1 in every 160 children in the world (0.63%) has an ASD, although the prevalence varies greatly depending on the country and its diagnostic capacity. In the case of Spain, the prevalence is 1%, similar to the European average, which means that there may be some 470,000 people with ASD [3].

This disorder presents a great variability in terms of its manifestations (hence the term "spectrum"), so it can be said that no two individuals with ASD are similar, since it is a heterogeneous disorder with manifestations that vary depending on the severity or level of involvement [4]. This variability is influenced, apart from the individual's specific development, by the support they may receive, the presence of comorbidity (the existence of other associated disorders or illnesses, such as psychiatric disorders or intellectual disability) [5,6], as well as the level of language development. This makes it necessary to approach it considering the specificity of each patient, and it is essential to offer individualized support adapted to the particular circumstances and abilities of each patient [4].

For people with ASD, dealing with certain environments and situations that are not familiar to them, can be a source of anxiety. Thus, for example, shopping in a supermarket or travelling with children with ASD can be very challenging for parents and very stressful for children [7]. Supporting these everyday situations is of interest to the scientific community, which is trying to offer solutions and mechanisms to help people with autism to manage these everyday activities [8–10]. Among all these day-to-day activities, our focus is on the support for attendance at medical appointments. In order to deal with this situation it is necessary for family members or caregivers of persons with ASD to prepare in advance for the visit and explain the details of the procedure to be performed at the medical consultation, with higher-level cognitive processes related to choosing, activating and maintaining different courses of action necessary for goal attainment [11].

A tool such as anticipation boards help to prepare the individual for an activity, identifying cognitive flexibility, included in executive functioning, as the most complex skill demanded during the process [12]. These boards allow us to break down the steps that make up an activity using pictograms and phrases. In this context, the use of technology can bring great benefits when it comes to acquiring new learning and enhancing developmental skills such as communication, autonomy, social interaction, etc. [13].

Although the use of information technology to support these users is increasing considerably [14], most of these systems do not take into account the specific characteristics that software for this population must support (visual processing, differentiation in sensory perception, attention to detail, predilection for a certain routine or order, as well as for certain sounds, objects and people, that is, adaptation to each user) [15]. For this reason, family members and caregivers often rely on handmade methods (paper, plasticizer, and self-adhesive tape) in order to develop systems to support planning and anticipation, as well as Augmentative and Alternative Communication Systems (AACS) (forms of expression other than spoken language that aim to increase the level of expression (augmentative) and/or compensate (alternative) for the communication difficulties that some people have in this area).

With the aim of providing an answer to the need of anticipation and communication identified in the population with autism, we have created a simple and intuitive app for tablets that allows parents or caregivers to plan the visit to the medical center, with the main objective of preparing and forecasting the situation (cognitive flexibility) and generate a less stressful situation for the child, due to the possible negative repercussions generated in the areas of functioning [16]. A User-Centered Design (UCD) process has been followed in the development of this application, which focuses on the end-user, their characteristics and

needs [17]. In addition, specific recommendations and guidelines for software oriented to users with ASD has been applied [18]. The ultimate goal is to improve the acceptance of the created software, its usability and the user experience (UX). Standard ISO 9241-11 defines usability as the "extent to which a system, product or service can be used by specified users to achieve specified goals with effectiveness, efficiency and satisfaction in a specified context of use". Standard ISO 9241-210 defines user experience as a "user's perceptions and responses that result from the use and/or anticipated use of a system, product or service". Additionally, this standard describes UX as "users' perceptions and responses include the users' emotions, beliefs, preferences, perceptions, comfort, behaviors, and accomplishments that occur before, during and after use".

The app, named PlanTEA, incorporates entertainment mechanisms during waiting times, positive reinforcement methods (rewards) and will facilitate, thanks to the inclusion of an augmentative and alternative communication booklet, communication with medical staff, when necessary. The usability and acceptance of the first version of this app has been evaluated by a set of stakeholders (although not including ASD users), which has led to very positive results as well as a set of proposals for improvement and extension of PlanTEA.

Although, in the past few years, there has been a proliferation of applications to support the ASD population, most of them focus on cognitive training or emotion recognition, with very few supporting the planning of everyday tasks. Those that support anticipation and planning do not usually allow the incorporation and categorization of pictograms or personalized images (representing objects, places or people known to the user), nor do they usually support multiple roles; and, in case they do, they are not free of charge. Furthermore, they do not incorporate alternative and augmentative communication mechanisms, nor do they integrate access to it in the planning itself; and most of them do not consider waiting times management or rewards. The application described in this article incorporates all these features. In addition, it is noteworthy that, compared to most of the existing developments in the literature, the design process of PlanTEA has followed specific design guidelines for this group of users, thus focusing not only on functionality but also on improving the usability of the developed application.

The paper is organized as follows: related works are discussed in Section 2; the PlanTEA system description is presented in Section 3; Section 4 describes the results obtained in a first evaluation of this app, and Section 5 addresses the discussion, including the main limitations. Finally, Section 6 presents a set of conclusions and further works.

## 2. Related Works: Technology and Autism Spectrum Disorder

The use of information and communication technologies (ICT) in interventions with people with ASD is increasing significantly [19], due to the great interest that technological devices raise in most of them. There is recent evidence on the convenience of using technological tools to develop and improve the communication and social skills of people with ASD [20], in particular, the use of mobile phones and tablets, as tools to improve academic, communication, occupational, leisure and independence skills in daily activities [15]. Among its advantages are the ability to provide visual information, personalized according to the communication needs of each individual, and their sensory difficulties, preferences and interests, as well as offering highly predictable environments [8]. In fact, numerous applications have been developed with the aim of enhancing and improving the quality of life of these users through the use of innovative technologies (serious games, wearables, robots, etc.) [21], transferring the learning acquired during the process to everyday life [22]. In particular, the use of game-based applications has grown considerably, especially when the target audience is children [23]. The use of this type of technology provides sensory information (mainly visual, auditory, and haptic) that allows them to interact with the simulated activity [24].

From the field of research on users with ASD [25], the use of ICT tools allows researchers to use appropriate practices aimed at the improvement and inclusion of skills

and abilities that present some kind of deficit in their daily lives [26]. Mainly it has focused on its use as a mediation of learning skills (e.g., social skills) or overcoming a problem. This type of users need environments that allow them not only to experience different situations but also to develop the ability to adapt to a problem (problem solving) [27].

In order to address this change, it is necessary to direct these activities to individual limitations and capabilities that allow the development of these strengths (problem-oriented approach) in flexible and individually adaptable learning environments that could foster the creativity of the participants. Thus, the technologies used could be dynamically modified and evaluated, allowing children to play an active role in the investigation [28]. Voos et al., through a randomized clinical trial, demonstrated the potential effectiveness of a behavioral intervention for children with ASD through the use of technologies [29]. The research was based on the emotional aspect of activities through the observed improvement in social behavior. It also points out the potential of digital therapy in the family environment to enhance cognitive aspects that are determinant in activities of daily living.

On the other hand, Parson et al. analyzed, in their research, the TOBY application that allows working cognitive aspects although with nuances, since it is not appropriate for all people with ASD, and should serve as a complement to other therapies and not in isolation [30]. It explains how the information provided by parents (family environment) allowed them to improve their children's abilities as well as serve as active agents of change in their development, congruent with the results of previous studies [31,32].

The study of related work has allowed us to observe that even though there is a growing number of software solutions to support people with ASD, most of the developments in this area are aimed at cognitive training and emotion recognition [19]. Support is usually provided through the use of mobile devices (smartphones and tablets), and many of these developments employ game mechanics (i.e., gamification techniques) as a mechanism for motivation and positive reinforcement [14,23,33]. However, many of these tools do not provide free access to all functionalities and do not include support for multiple user roles. Although the usefulness of mobile applications in promoting the delivery of health care, health-promoting habits and behaviors is increasingly recognized [34], there are hardly any applications that enable children with ASD to manage the phase of preparation or anticipation in people with ASD when they go to their doctor for a regular appointment, and its usefulness has not been evaluated. One of the main reasons why it is difficult to find scientific evidence on the subject is the difficulty encountered by researchers, either because of the limitations of use, the lack of adherence to the intervention or the great heterogeneity of the pathology itself.

## 3. PlanTEA: System Description

With the aim of supporting the need for anticipation and communication for children with ASD in a necessary activity in their lives (attending medical appointments), the PlanTEA application was developed for mobile devices tablet type, on the Android operating system. PlanTEA allows planning medical visits and medical examinations for children with ASD between 4 and 10 years old, as these are unavoidable tasks that can generate a lot of anxiety for them and, therefore, for their families. At the moment, the user interface of the app is in Spanish, although multi-language support will be provided in future versions.

PlanTEA (Figure 1) includes two main functionalities (Figure 1a): a planning and anticipation system, and an augmentative and alternative communication system (AACS); both based on the use of pictograms, which makes their use possible for both verbal and non-verbal children with ASD. This app has two main contexts of use: the home and the medical center. At home (Figure 1b), parents or caregivers can prepare and plan the steps of the medical appointment, defining the rewards and activities that the child can perform during the waiting time. The planning will be done using pictograms provided in the tool and related to the medical domain (places, people, objects, and common actions in medical appointments), but it will also be possible to import external resources from the internet or

using the device's camera (Figure 1d). At home, the child can preview the plan as many times as needed to know what to expect. The other scenario of use is at the medical center (Figure 1c), where the child can follow the plan and communicate via the AACS with the medical professionals (he/she can communicate symptoms, their duration, intensity, as well as the parts of the body affected).

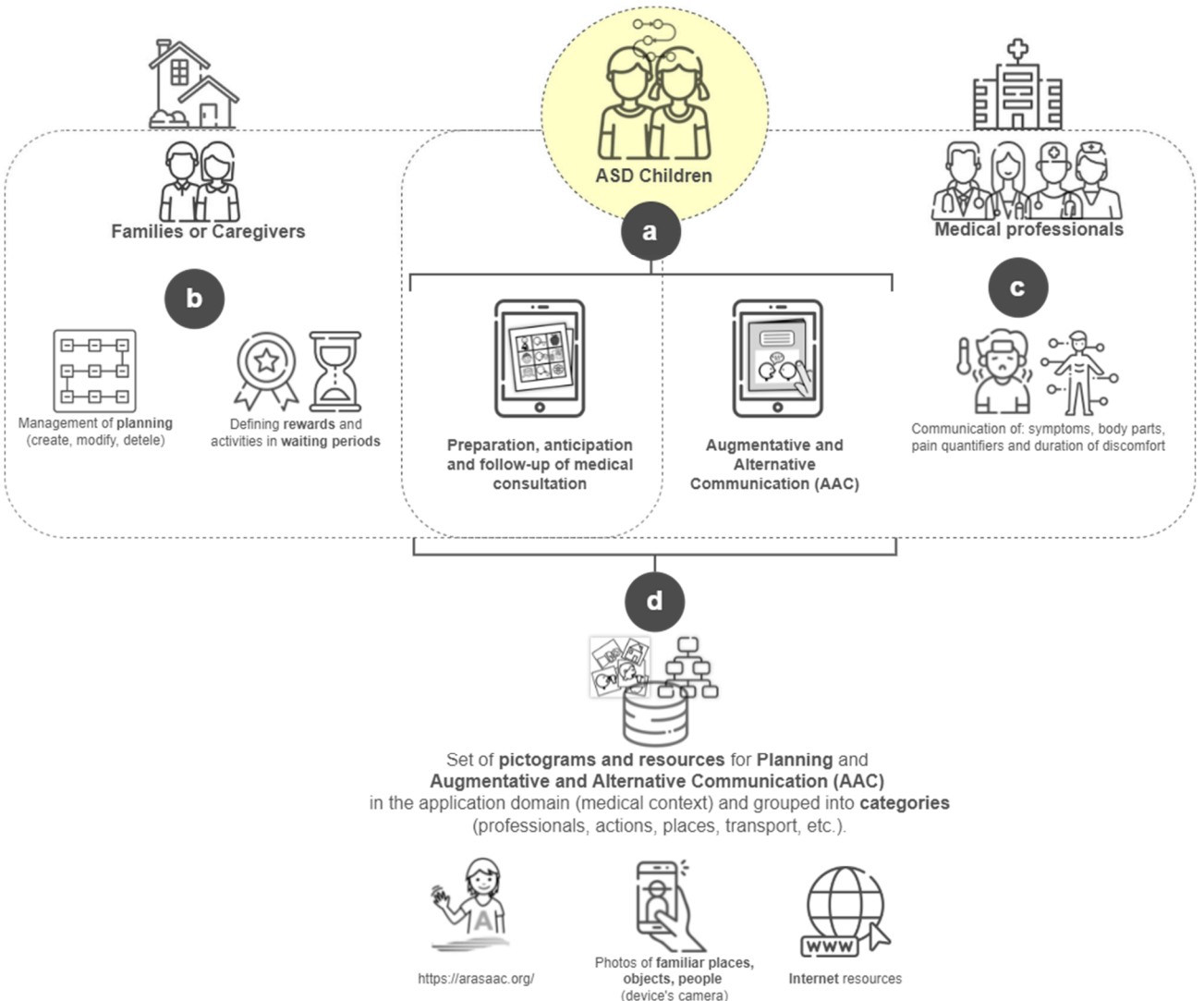

**Figure 1.** Overview of PlanTEA. (**a**) Functionalities available for children with ASD; (**b**) Functionalities available for family members and caregivers; (**c**) Functionalities available to the staff of medical centers; (**d**) Sources of pictorial information and resources.

### 3.1. Development Process of PlanTEA

According to the User-Centered Design approach, the best designed products and services result from understanding the needs of the people who will use them. Therefore, in the requirements specification and design process of PlanTEA, a UCD approach was followed, paying special attention to aspects such as the use of visual metaphors and the elimination of distractions. A participatory design process (Figure 2) was followed, that involved the active participation of experts in mobile application usability as well as in the domain of the application, i.e., experts in occupational therapy with people with ASD, as well as family members of children who suffer from this disorder (Figure 2a).

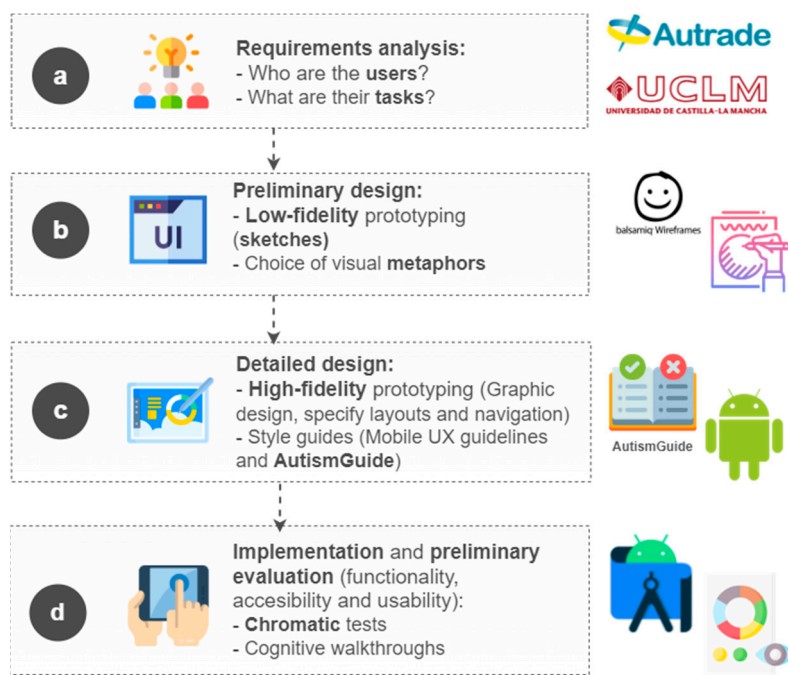

**Figure 2.** Overview of the PlanTEA's development process. (**a**) Requirements analysis; (**b**) Preliminary design; (**c**) Detailed design; (**d**) Implementation and preliminary evaluation.

As for the design phase of the app, in its earliest stage (Figure 2b), low-fidelity prototypes were created using Balsamiq Wireframes tool (https://balsamiq.com/wireframes/ (accessed on 1 April 2022)), evaluated and refined in successive iterations through the participation of families and specialists. In the process of designing the graphical interface (Figure 2c), special care was given to aspects related to the usability and accessibility of the application, as well as the characteristics of the two roles supported (parents/caregivers and children with ASD with verbal and non-verbal communication). Meanwhile, the detailed design phase of PlanTEA's user interface took into account the recommendations or usability guidelines (AutismGuide) proposed by Aguiar et al. [18], considering those that were applicable to the specific objective of the application developed (planning and communication) (Table 1).

**Table 1.** Recommendations on usability to design software solutions for users with ASD (AutismGuide).

| AutismGuide Recommendation | "Software Solutions Designed for Users with ASD Must . . . " | PlanTEA Support * | Explanation and Proposals for Improvement |
|---|---|---|---|
| General Usability Principles | . . . be useful (accomplishing its original purpose), efficient (performing as well as possible) and functional (works properly, no malfunctions, no bugs). | √ | PlanTEA is considered to be efficient and well-functioning. In any case, this will be measured by several evaluations using acceptance, usability, and usefulness frameworks (a preliminary evaluation with part of the stakeholders is described in Section 4). |
| Nonfunctional Requirements | . . . have long lifespan and high availability, same appearance as devices available to the general public, large screen, good-quality camera, . . . | ~ | The app is designed specifically for use on tablet devices. Smartphone adaptability needs to be improved. |

**Table 1.** *Cont.*

| AutismGuide Recommendation | "Software Solutions Designed for Users with ASD Must …" | PlanTEA Support * | Explanation and Proposals for Improvement |
|---|---|---|---|
| Functional Requirements | … have a profile restricted to practitioners, parents and caregivers so that they can manage access. | √ | PlanTEA has two roles: relative or caregiver and child. The access to the profile of the caregiver or family member is done through a password. |
| Adaptability | … react to context and these users' needs and preferences. | ~ | PlanTEA takes into account the particular interests of the children (rewards, activities during waiting times) and the personalization of the media (images of objects, places and familiar people), using the device's camera or by downloading them from the internet. However, there are areas for improvement: (a) incorporating planning in verbal formats; (b) possibility of customizing the appearance of the controls and texts that compose the interface (position, size, colors, font, etc.) of the GUI according to the sensory particularities of the users and their preferences. |
| Guidance | … provide assistance in the form of messages, alerts, icons, layout, etc. | √ | The app has a logical, simple and clear layout. The navigation structure is also very simple, with the possibility to go back or return to the home screen, as well as the possibility to consult the help at any time (in text, image and video format). Task progress is indicated visually (e.g., by shading when progressing through the planning, or clearly recognizable visual feedback when certain milestones are reached). |
| Workload | … promote their perception, concentration, attention, memory, etc. | ~ | PlanTEA provides a clear, simple and minimalist interface (soft colors, easily distinguishable color codes to highlight key elements) and includes a small number of available functionalities. However, there are aspects whose effect we want to test: whether it is better to use plain colors for the backgrounds as well as the effect of the animations that are triggered when the child reaches a milestone in the planning. |
| Compatibility | … take account of their various characteristics (habits, skills, age, expectations, etc.) and adapt the tasks, navigation, layout, etc. accordingly. | ~ | The current version of PlanTEA targets children between 4 and 10 years old, supporting both verbal children and those who have not developed language (non-verbal), thanks to the use of pictograms. We are planning to broaden the spectrum of possible users to other ages and abilities (for example, to consider adults or adolescents with high-functioning autism, as well as to deepen in aspects related to the user's gender). |

**Table 1.** *Cont.*

| AutismGuide Recommendation | "Software Solutions Designed for Users with ASD Must ... " | PlanTEA Support * | Explanation and Proposals for Improvement |
|---|---|---|---|
| Explicit control | ... ensure that they always have control (e.g., pause, restart) over the computer processing. | √ | The app allows the user to select and interact with the interface elements and to navigate freely through the application. None of the supported tasks have expiry times and there is no automatic redirection between screens. It also supports the visualization of planning and the restarting of schedules as many times as desired. |
| Significance of Codes | ... display codes (icons, images, terms, etc.) that are easy for these users to understand. | √ | PlanTEA makes use of visual representations (pictograms, images, icons, pictures, and symbols) that are easy to understand. It also allows the inclusion of personalized images that are familiar to the child. |
| Error Management | ... allow them to avoid or reduce their errors. | ~ | The application avoids and repairs errors that the user may make, and when an action is impossible, it clearly indicates it. However, in future versions of PlanTEA we want to deepen and improve aspects related to the assistance and visual feedback that the application should provide in case of error: including help in multimedia format or using icons that represent facial expressions. |
| Consistency | ... maintain the same types of interface, navigation and interaction elements throughout. | √ | In the app design, consistency in terms of layout and the navigation and interaction elements used has been maintained. However, this aspect should be empirically assessed in future usability tests with end-users (children with ASD). |

* The degree of compliance with the recommendation in PlanTEA is indicated by the symbols: √ (supported) and ~ (partially supported).

In relation to the final phases, of implementation and preliminary evaluation, it is necessary to comment that the application was developed for Android-based tablet devices, and Android Studio was used (Figure 2d). The application was subjected to formative evaluations to check the functional and mobile usability aspects of the application. Regarding accessibility, chromatic tests were applied, and special care was taken with the size and location of the controls that made up the interface to facilitate selection by users who may have associated motor problem.

*3.2. Roles and Functionalities*

As mentioned above, the application supports two types of users or roles: *planner*, for family members or caregivers, in charge of designing the planning and managing the plans; and *child*, who will be able to view the selected planning and access the communication notebook.

PlanTEA supports the following set of specific functionalities:

- Planning the sequence of actions to be carried out before, during and after the medical appointment, making use of the so-called anticipation boards (Figure 3a).
- Management of the plans, allowing them to be stored for later use, modification, and deletion.

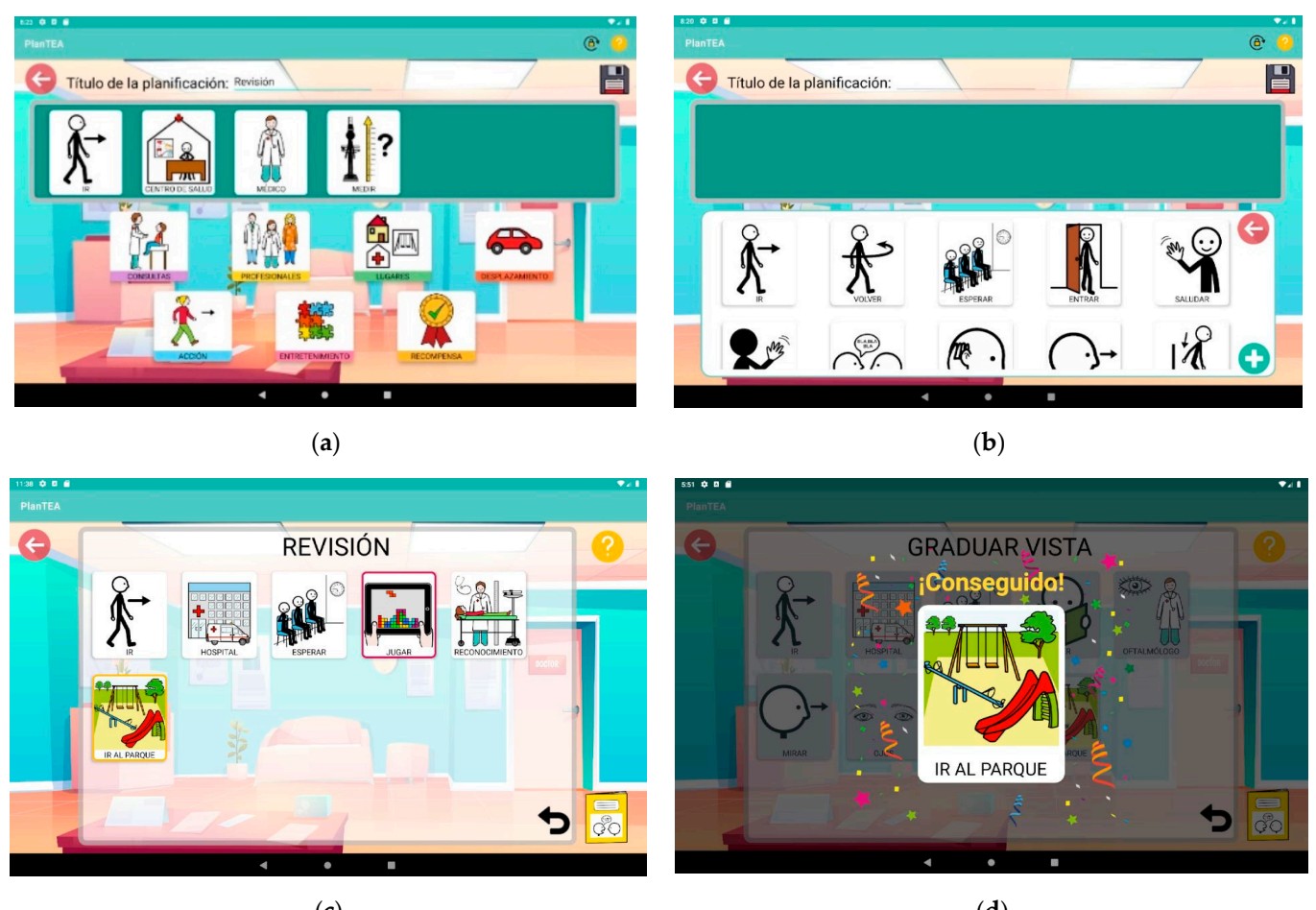

**Figure 3.** Appearance of the PlanTEA graphical user interface to support planning and anticipation. (**a**) Appearance of the graphical user interface of the planning design functionality (*planner* role). (**b**) Examples of pictograms available in the "action" category (*planner* role). (**c**) Functionality of review and/or follow-up of a planning (*child* role). (**d**) Achievement of a reward by the child (*child* role).

The planning will be based on the use of pictograms with the specific vocabulary of this domain (medical environments) (retrieved from the repository of "Symbol set and resources for Augmentive and Alternative Communicación (AAC)" de ARASAAC (Aragonese Center of Augmentative and Alternative Communication): https://arasaac.org/ (accessed on 1 April 2022)), as well as own images, uploaded from the gallery of the mobile device. This functionality allows real photographs of people, objects or places known to the child to be included in the planning. The available pictograms and images are grouped into categories (actions, places, people, types of medical consultations, entertainments during waiting times, rewards, among others) (Figure 3b).

- Management of waiting times that may be generated during the visit (both expected and incidental), through entertainment activities defined by the parents (playing, reading, etc.).
- Follow-up, by the child with ASD, of the selected plan (Figure 3c). As indicated above, the app allows for the inclusion of entertainment (to better cope with waiting times), as well as rewards or incentives at specific times in the planning (at times when parents or caregivers consider it most appropriate) (Figure 3d).
- Incorporation of a communication mechanism (Augmentative and Alternative Communication or boards) between the child and the doctor, when necessary (non-verbal ASD children or those with oral communication difficulties). In this section the child

can select from a set of available pictograms: symptoms (Figure 4a), body parts (Figure 4b), intensity and duration of pain or discomfort, among other aspects.

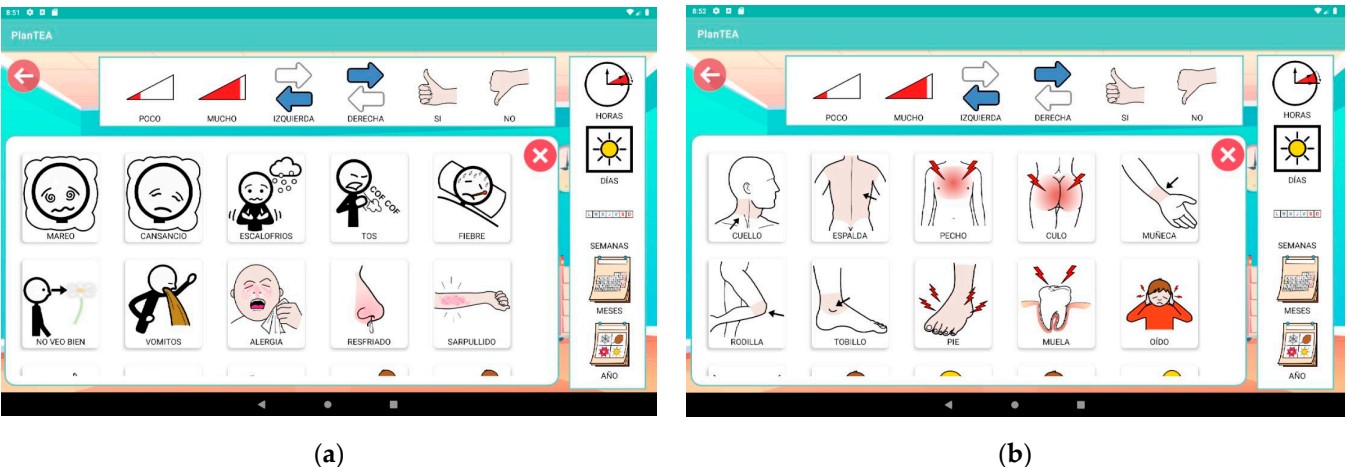

(**a**)        (**b**)

**Figure 4.** Appearance of the PlanTEA graphical user interface supporting augmentative and alternative communication. (**a**) Part of the communication booklet allowing the selection of symptoms. (**b**) Part of the communication booklet allowing selection of body parts.

## 4. A First Evaluation of PlanTEA

At the end of the development of the application, an evaluation of its usability and acceptance by experts (occupational therapists and experts in the design of interactive systems) and relatives of children with ASD was carried out aiming to check whether PlanTEA fulfils the expectations and requirements defined, and to detect possible shortcomings and opportunities for improvement.

### 4.1. Evaluation Design

In order to carry out an initial summative evaluation, a descriptive cross-sectional study was designed (Figure 5), based on the use of a subjective perception questionnaire [35] in which experts in different areas participated: family members of children with ASD, therapists or specialists in ASD, and applications developers. Given the social distance restrictions imposed by the COVID-19 pandemic, participants received by e-mail a link to a video demonstration of how PlanTEA works, together with other link to a questionnaire deployed with Microsoft Forms (https://www.microsoft.com/es-es/microsoft-365/online-surveys-polls-quizzes (accessed on 1 April 2022)), which they had to fill out after being sure that they had understood PlanTEA's functioning. The participation was voluntary, anonymous and all of them gave informed consent to take part in the study.

The measure instrument consisted of 12 items, with five levels of Likert scale responses, according to the degree of agreement or disagreement with each item (score of 5 represents "strongly agree" and 1 represents "strongly disagree"), and three open-ended questions. The first 8 items were based on the TAM (Technology Acceptance Model) framework [36], which measures aspects related to the usefulness, ease of use and intention to use a software system, and the remaining 4 items correspond to a reduced version of the SUS scale (System Usability Scale) [37], which evaluates the usability of an interactive system. The open questions allow the user to indicate strengths and weaknesses of the tool, as well as any other comments or suggestions for improvement.

The evaluation instrument also included several questions to characterize the participant, such as gender, profile (family member or caregiver of child with ASD, therapist or person with knowledge in ASD, software developer) and interest in being informed of future improvements or evolutions of the application.

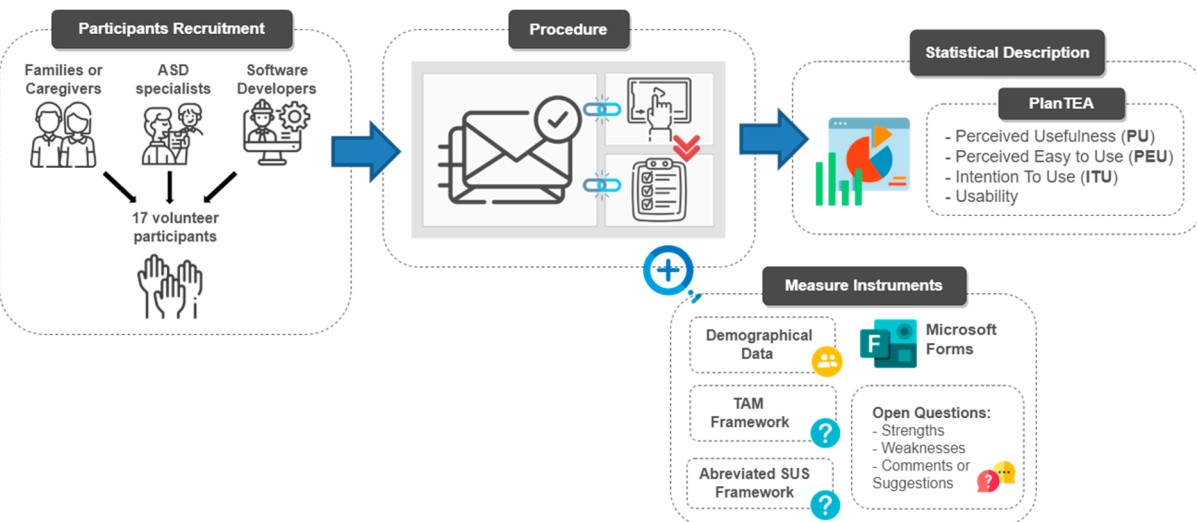

**Figure 5.** Design of the PlanTEA evaluation experience.

### 4.2. Participants

The video-demo of PlanTEA and the measure instrument was sent to 6 software developers with expertise in usability and 6 psychologists, technicians and therapists with expertise in ASD from the Spanish Federation of Autism (FESPAU), from the Regional Association (from the Castilla-La Mancha region in Spain) of People with Autism and Other Developmental Disorders (AUTRADE), from a Child Development and Early Attention Centre (CDIAT Fuente Agria in Puertollano), as well as to 5 families/caregivers of ASD children connected to these associations. There was a balanced participation of men (53%) and women (47%).

### 4.3. Results

Table 2 shows the descriptive statistics (mean, median and mode), as well as the percentages of responses to the different items of the TAM framework: Perceived Ease of Use (PEU), Perceived Usefulness (PU) and Intention To Use (ITU).

**Table 2.** Descriptive statistical results and response percentages related to the acceptance of the tool (questions based on the TAM framework).

| Questionnaire | | Likert Scale | | | | | | | |
| --- | --- | --- | --- | --- | --- | --- | --- | --- | --- |
| | | Descriptive Statistics | | | Percentages | | | | |
| ID | Statement | Mean | Median | Mode | 1 | 2 | 3 | 4 | 5 |
| PU1 | Using PlanTEA allows to plan, quickly and efficiently, the sequence of steps to be followed in a medical appointment | 4.76 | 5.00 | 5.00 | - | - | - | 23.5% | 76.5% |
| PU2 | I find PlanTEA to be useful for improving communication of people with ASD in medical contexts | 4.76 | 5.00 | 5.00 | - | - | 5.9% | 11.8% | 82.4% |

**Table 2.** *Cont.*

| Questionnaire | | Likert Scale | | | | | | | |
| | | Descriptive Statistics | | | Percentages | | | | |
| ID | Statement | Mean | Median | Mode | 1 | 2 | 3 | 4 | 5 |
| PU3 | I believe that PlanTEA would help to anticipate and deal better with waiting times in medical contexts for people with ASD | 4.41 | 5.00 | 5.00 | - | 5.9% | - | 41.2% | 52.9% |
| PEU1 | I find the interaction with PlanTEA to be flexible. | 4.59 | 5.00 | 5.00 | - | - | - | 41.2% | 58.8% |
| PEU2 | I find PlanTEA easy to use | 4.82 | 5.00 | 5.00 | - | - | - | 17.6% | 82.4% |
| ITU4 | I would like to use PlanTEA | 4.65 | 5.00 | 5.00 | - | - | - | 35.3% | 64.7% |
| ITU2 | I intend to use PlanTEA in the future | 3.71 | 4.00 | 5.00 | 11.8% | - | 29.4% | 23.5% | 35.3% |
| ITU3 | I would recommend using PlanTEA | 4.76 | 5.00 | 5.00 | - | - | 5.9% | 11.8% | 82.4% |

Analyzing the results, it can be observed that the best average corresponds to the question "I find PlanTEA easy to use" and the worst average for the question "I intend to use PlanTEA in the future". On the other hand, 82.4% of the participants strongly agree that the application is useful to improve communication in medical contexts, easy to use and recommendable.

Table 3 shows the results obtained for the SUS scale items. The evaluators agreed that PlanTEA is an accessible, simple and easy-to-use. The best average corresponds to the question "I believe that the different functions of PlanTEA are well integrated and are easily accessible". Furthermore, 82.4% of the participants think that no help from a person with technical knowledge is required to use the application.

**Table 3.** Descriptive statistical results and response percentages related to usability of the tool (questions based on the SUS scale).

| Questionnaire | | Likert Scale | | | | | | | |
| | | Descriptive Statistics | | | Percentages | | | | |
| ID | Statement | Mean | Median | Mode | 1 | 2 | 3 | 4 | 5 |
| SUS1 | I think I would need help from a person with technical knowledge to use PlanTEA | 1.18 | 1.00 | 1.00 | 82.4% | 17.6% | - | - | - |
| SUS2 | I believe that the different functions of PlanTEA are well integrated and are easily accessible | 4.76 | 5.00 | 5.00 | - | - | - | 35.3% | 64.7% |
| SUS3 | I think most people would learn to use PlanTEA quickly | 4.53 | 5.00 | 5.00 | - | - | - | 47.1% | 52.9% |
| SUS4 | I think I would need to learn many other things before I would be able to use PlanTEA properly | 1.53 | 1.00 | 1.00 | 76.5% | 11.8% | - | 5.9% | 5.9% |

According to the groups of participants, Table 4 shows the results obtained for the items and dimensions of the questionnaire. The main finding is that the mean of all groups in the different dimensions is similar, with only the therapists scoring lower in the PU dimension and the developers scoring lower in the ITU2 item. Both results are logical: in the first case, because therapists are more aware of the difficulties that children with ASD have; in the second case, because developers probably do not have family members with ASD, so they do not intend to use the application in the future.

**Table 4.** Descriptive statistical results according to the different roles of participants.

| Item | Familiar | | Software Developer | | ASD Therapist | |
|---|---|---|---|---|---|---|
| ID | Mean | St Dev | Mean | St Dev | Mean | St Dev |
| PU1 | 5.00 | 0.00 | 5.00 | 0.00 | 4.33 | 0.52 |
| PU2 | 4.80 | 0.45 | 5.00 | 0.00 | 4.50 | 0.84 |
| PU3 | 4.40 | 0.55 | 4.67 | 0.52 | 4.17 | 1.17 |
| **PU** | **4.73** | **0.28** | **4.89** | **0.17** | **4.33** | **0.70** |
| PEU1 | 4.60 | 0.55 | 4.50 | 0.55 | 4.67 | 0.52 |
| PEU2 | 4.80 | 0.45 | 4.83 | 0.41 | 4.83 | 0.41 |
| **PEU** | **4.70** | **0.27** | **4.67** | **0.41** | **4.75** | **0.42** |
| ITU1 | 4.60 | 0.55 | 4.83 | 0.41 | 4.50 | 0.55 |
| ITU2 | 4.20 | 0.84 | 2.83 | 1.60 | 4.17 | 0.98 |
| ITU3 | 4.80 | 0.45 | 5.00 | 0.00 | 4.50 | 0.84 |
| **ITU** | **4.53** | **0.51** | **4.22** | **0.54** | **4.39** | **0.74** |
| SUS1 | 1.00 | 0.00 | 1.33 | 0.52 | 1.17 | 0.41 |
| SUS2 | 4.80 | 0.45 | 4.50 | 0.55 | 4.67 | 0.52 |
| SUS3 | 4.40 | 0.55 | 4.67 | 0.52 | 4.50 | 0.55 |
| SUS4 | 1.60 | 1.34 | 1.00 | 0.00 | 2.00 | 1.55 |
| **SUS** | **2.95** | **0.48** | **2.88** | **0.14** | **3.08** | **0.47** |

In addition, most participants (76%) showed interest in staying informed about the evolution of the application.

Finally, open answers provided other relevant results about PlanTEA. Some of the comments received regarding the strengths of the application are:

- "Simple and easy. Very intuitive and understandable".
- "It is very straightforward and well-focused".
- "Intuitive, practical, doesn't take too much time to plan".
- "Well adapted to the needs of people with ASD, user-friendly design and ease of use".
- "Flexibility in generating sequences, inserting new pictograms as well as images from device memory".
- "The fluid and natural interaction when creating the plans, which avoids having to create the pictograms "by hand" and spending a lot of time preparing such a plan manually. The possibility of including "personalized" pictograms of places, places and people known to the child, using the device's own camera, or locating them in any external repository".

The comments received highlight the usefulness of the tool developed, indicating that it is also flexible and usable, i.e., easy to use and learn by potential users. The possibility of including external resources (from the Internet or using the device's camera) is also highly valued. The possibility of including customized and familiar content is a particularly useful feature for people with ASD.

In terms of weaknesses of the tool indicated by participants, the following are listed:

- "The possible overloading of some screens of the app, in particular in the communication notebook, where the inclusion of so many icons or pictograms on the screen at the same time can act as a distractor and interfere in the task".

- "The fact that the app is aimed only at children. It would be interesting to broaden the age range of the target population, as the app can be useful for both children and adults".

In terms of areas for improvement or weaknesses, several respondents insisted on the desirability of adapting the application for use by adults. The complexity or overloading of some screens is also an aspect criticized by participants.

Regarding comments and suggestions for improvement, the following can be highlighted:

- Proposals by ASD experts and therapists:
  - Include pictograms/photographs of common objects in the consultation room such as a phonendo or syringe, in such a way that what they are used for can be shown.
  - Include a complete human figure rather than separate body parts to indicate areas of pain.
  - Consider the use of alternative representations (metaphors) for the concepts "too much/too little". For example, it is proposed to represent the degree of pain using the metaphor of the thermometer.

- Proposals for improvement by respondents with a more technical profile and usability experts:
  - Include a search engine to make it easier to locate pictograms, rather than navigating through the different categories.
  - Integrate the application with a third-party calendar and OAuth service (Google Calendar, Outlook, etc.).
  - In relation to the planning functionality, include the possibility to access public pictogram databases and search by keywords.

- Proposals for improvement by family members and caregivers:
  - Include multimedia content ("animated tours" or videos) of the steps to be followed in consultations. Displaying and reviewing these videos and animated representations will improve and better cover the anticipation needs of users.
  - In relation to the rewards, include the possibility, through the app, to directly access games included in the planning and installed on the device, so that the child can play while waiting in the consultation room.

### 4.4. Limitations

There are several threats to the validity [38] of the study that could have influenced the results obtained. Given that the evaluation made is purely descriptive, the statistical conclusion validity is the threat that least affects this study. In this sense, we have tried to enhance our results by the proper application of descriptive statistical tests and to provide the user with no dichotomic variables to avoid the restriction of range.

In order to guarantee construct validity, attention must be paid to the formulation of the items and the length of the scale [39]. Thus, the measure instrument was based on two validated frameworks, including quantitative and qualitative items, since responses to open-ended questions provide relevant information. An attempt was made to use unambiguous vocabulary when translating items from English to Spanish. However, the reliability of the scale has not been calculated, what may limit the validity of the construct. This thread should be addressed in future studies.

In terms of threats to internal validity, the use of subjective surveys may cause participants to give biased answers. This may be mainly due to two reasons [40]: because they lacked the necessary knowledge to properly answer the questions or because the results are based upon the perceptions of the respondents. Regarding the first one, we have tried to make the demonstration video as explanatory as possible. However, for future evaluations, we plan to provide supplementary material such as a detailed written manual. Concerning the second reason, some uncontrolled extraneous factors, such as personal feelings or circumstances, the day of the week or the time of day, the voluntariness of participation, its motivation, etc. that could affect participants' responses are beyond our reach.

To guarantee the external validity of empirical studies, it is recommendable to recruit representative participants of the target population. The results obtained in this preliminary evaluation are obtained from a sample formed by relatives of children with ASD, therapists or specialists in ASD, and software developers. Given the profile of the participants in the experiment, their voluntariness, as well as the small number of them, the representativeness and generality of the results is rather questionable. This is a recurrent problem in research targeting users with ASD. Most of the existing developments in the literature are either not evaluated [41] or are evaluated with small samples [42]. As indicated in the discussion section, the next step is to evaluate PlanTEA by users with ASD.

## 5. Discussion

Despite the limitations described, the experience carried out to evaluate this prototype has yielded very positive and promising results. It has also served to detect certain aspects that could be improved and to identify possible extensions or improvements to the system.

Among the extensions of PlanTEA that are planned to be addressed, the extension of the range of users that could make use of its planning functionality stands out. In addition to verbal and non-verbal children, there is a group of people with Autistic Disorder (AD) who have been traditionally forgotten by research: children or adults with High Functioning Autism (HFA) [43]. The term HFA, although not being officially recognized in the two most used classification systems at international level (DSM or ICD-10), is commonly used to identify patients diagnosed with AD or PDD-Not Other Specified (PDD-NOS), with average or above average intellectual abilities (Intelligent Quotient, IQ, higher than 70) [44]. Those persons carry the main characteristics of autism, although attenuated, in relation to the indicators provided by the main measurement and diagnosis instruments used.

The fact that people with HFA can have an academic performance and other cognitive indicators at similar levels to those of the neurotypical population, and even in some cases higher, can lead to underestimate the needs and difficulties that these people can have in their daily life (shopping, using public transport, etc.). Although the signs of autism may not be so evident, these people continue presenting problems of communication and interpretation of non-verbal signals, social interaction, application of rules and social protocols, as well as presenting behavioral rigidity, needing routine and order. Everyday situations can sometimes be overwhelming for them, causing increased anxiety and stress. In addition, they are often the victims of teasing and bullying, which has a negative impact on their self-confidence and self-esteem and makes them prone to explosive, defiant and oppositional attitudes, isolation, etc.

Therefore, in the next version of the PlanTEA app (PlanTEA 2.0), several modifications and extensions are proposed:

- Incorporate as a user profile people with HFA, who do not necessarily need to make use of pictograms when planning and/or following a plan. In other words, support for the creation, editing and consulting of plans in verbal format.
- In relation to the previous issue, give the possibility to the family member or caregiver, taking into account the characteristics of the user, to choose which type of communication is the most appropriate: "text only", "image only" or "text + image".
- Extend the application domain (attending medical consultations) and generalize its use to other scenarios or tasks (shopping at the supermarket, attending job interviews, etc.).
- Improve visual time feedback in schedules. This will allow users to better manage time and waiting times (visual indicators of start, end, progress, duration, timetable, etc.).
- Incorporate long-term planning functionality by including calendars, notifications and reminders.
- Incorporate functionality for the definition and monitoring of "social stories". A social story is a short, individualized narrative that is used to clarify difficult or confusing situations for people with autism. More specifically, a social story is written to provide information about what people in a given situation think or feel. They represent a series of experiences reflecting the social cues and their importance, and the script of

what should or can be done and said, in other words, the "what", "when", "who" and "why" of social situations. Social story interventions [45], delivered via a tablet device, have been shown to improve autonomy, reduce anxiety in their daily lives, and allow them to self-empower themselves without the help of adults [46].

- Incorporate mechanisms for self-direction and self-regulation during planning (e.g., incorporate reminders to carry a "reassuring object or item").
- Support the internationalization of the app. Currently the application only supports the Spanish language.

In addition to the set of improvements listed above, some aspects related to the visual metaphors used (e.g., to indicate degree of pain), the amount of distractors or the appropriateness of the use of "rewards" (positive reinforcement) in the case of adult users will be researched and reviewed. In the process of redesigning the graphical interface of the application to adapt it to the type of user, who may be a child or an adult (HFA), specific aspects of usability [18] and the user experience (UX) of users with ASD will be taken into account [47].

We also consider it necessary to improve the management and search for pictograms and images to be included in the plans. In this respect, it should be interesting to use Artificial Intelligence techniques, based on a Deep Learning or a Machine Learning approach, to navigate and manage pictorial content in a more automated way considering the user role and his/her needs [48,49].

Once the changes and extensions mentioned above have been made, an evaluation of PlanTEA 2.0 will be carried out with end-users (children and adults with HFA), who are members of the AUTRADE association. This evaluation will follow the user experience evaluation methodology for people with ASD proposed by [50].

## 6. Conclusions and Future Works

Performing everyday tasks, such as attending medical appointments, can be a challenge and a source of stress and anxiety for people with ASD and their relatives or caregivers. Anticipating and planning these situations is a great help for all of them, for which the use of software applications for smartphones or tablets can play an essential role.

To this end, an application for Android tablets (called PlanTEA) has been developed to anticipate and plan medical appointments for children with ASD. The application incorporates entertainment mechanisms for waiting times, positive reinforcement methods, and facilitates communication (augmentative and alternative) with medical staff. This app has undergone a first evaluation with experts in mobile usability and autism, as well as families/caregivers of people with ASD, leading to very positive results and a list of future improvements and extensions. Among them, the extension of its use to a wider range of everyday situations (taking a means of transport, shopping in a supermarket, etc.) and to a wider spectrum of user, including people with high-functioning autism, stands out. Therefore, one of our future lines of work is to consider this group of potential PlanTEA users.

On the other hand, one of the main shortcomings of the preliminary evaluation conducted is related to the profile of the participants, which does not include children with ASD. Therefore, a more comprehensive evaluation of PlanTEA is the other main line of work to be continued. In this sense, it is intended to increase the sample of participants and the representativeness of the two supported roles, as well as to apply of an evaluation methodology that considers specific aspects of usability and user experience of people with ASD.

**Author Contributions:** Conceptualization, A.I.M., C.L. and P.H.; methodology, A.I.M., C.L., A.T.-G., P.H. and C.R.; investigation, A.I.M., C.L., P.H. and A.T.-G.; writing—original draft preparation, A.I.M.; writing—review and editing, A.I.M., C.L., A.T.-G., P.H. and C.R.; supervision, A.I.M., C.L. and A.T.-G. All authors have read and agreed to the published version of the manuscript.

**Funding:** This research received funding from "UCLM-Telefónica Chair in Advanced Interaction Systems for Digital Education" and European Regional Development Funds (FEDER): 2018/11744.

**Institutional Review Board Statement:** The users' participation was voluntary. We keep the users' identity as anonymous. The experiment was not intrusive or affected the participants in any sense.

**Informed Consent Statement:** Informed consent was obtained from all subjects involved in the described evaluation.

**Data Availability Statement:** The data presented in this study are available on request from corresponding author.

**Acknowledgments:** We would like to thank, first of all, the student Sara Lara Caro who developed the PlanTEA application in the context of her final degree project. We would also like to thank the Telefónica-UCLM Chair for providing financial support to this project and, finally, the participants from different groups (specialists, family members and technical experts), who voluntarily participated in the preliminary evaluation of PlanTEA described in this paper.

**Conflicts of Interest:** The authors declare no conflict of interest.

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
