# Peer review of "PlanTEA: Supporting Planning and Anticipation for Children with ASD Attending Medical Appointments"

_applsci, doi:10.3390/app12105237_

Round 1

Reviewer 1 Report

The paper entitled "PlanTEA: Supporting planning and anticipation for children with ASD attending medical appointments" presents PlanTEA, a new software tool that allows the planning of the attendance of children with ASD to the necessary medical appointments, and their communication with specialists. A preliminary evaluation from stakeholders is also reported with good acceptability.

The utility of such an application is highly needed for autistic people and their families, and the paper is overall clear and well-written. My comments mainly regard the terminology used and the description of the autistic condition.

  • "Autism spectrum disorder (ASD) is defined as a chronic neurological dysfunction with": ASD is a neurodeveopmental disorder, not a neurological dysfunction
  • In general, more nuanced terms should be used in this first part of the paper
  • Line 43: cognition is not always impaired in people with ASD, therefore I suggest to delete this word
  • Line 46: again, ASD is not a neurological condition but a neurodevelopmental condition
  • Line 57-60: the severity of the disorder also depends on the medical comorbidities, see for instance the paper by Brondino et al. (doi: 10.1007/s11606-019-05071-x) that are also directly related to the utility of the PlanTEA App
  • Line 202: the recommendations or usability guidelines (AutismGuide) proposed by *the authors are missing here* [13]
  • In the discussion there is mention to "high-functioning autism", which is not really accepted by a part of the autistic community. The authors should consider changing this expression.

Author Response

We are grateful for the reviewer's comments and suggestions for changes that will improve the submitted article. All changes in the manuscript have been highlighted in blue.

The paper entitled "PlanTEA: Supporting planning and anticipation for children with ASD attending medical appointments" presents PlanTEA, a new software tool that allows the planning of the attendance of children with ASD to the necessary medical appointments, and their communication with specialists. A preliminary evaluation from stakeholders is also reported with good acceptability.

The utility of such an application is highly needed for autistic people and their families, and the paper is overall clear and well-written.

My comments mainly regard the terminology used and the description of the autistic condition.

  • "Autism spectrum disorder (ASD) is defined as a chronic neurological dysfunction with": ASD is a neurodeveopmental disorder, not a neurological dysfunction

Authors> We have corrected this error along the paper taking into account the reviewer's suggestion.

  • In general, more nuanced terms should be used in this first part of the paper

Authors> We have considered this suggestion trying to use more nuanced terms in the Introduction section.

  • Line 43: cognition is not always impaired in people with ASD, therefore I suggest to delete this word

Authors> We agree with the reviewer, so we have removed the word "cognition", leaving only communication, socialization and reciprocal interaction as characteristic features of people with ASD.

  • Line 46: again, ASD is not a neurological condition but a neurodevelopmental condition

Authors> It has been modified in the text.

  • Line 57-60: the severity of the disorder also depends on the medical comorbidities, see for instance the paper by Brondino et al. (doi: 10.1007/s11606-019-05071-x) that are also directly related to the utility of the PlanTEA App

Authors> Thanks to the reviewer for providing this relevant reference, which has been included in the article.

  • Line 202: the recommendations or usability guidelines (AutismGuide) proposed by *the authors are missing here* [13]

Authors> It has been corrected.

  • In the discussion there is mention to "high-functioning autism", which is not really accepted by a part of the autistic community. The authors should consider changing this expression.

Authors> Certainly, "high-functioning autism" is not an official medical term, but an informal term commonly used in this field. Therefore, we have decided to keep it in the article, although making its definition clear and supporting its use in several bibliographical references in which the use of this term is justified.

Reviewer 2 Report

This paper presents an IT approach for managing children with ASD as it concerns their QoL and the continuous communication with their environment, informal carers and physicians.

The topic is interesting, but it seems that this paper shows a preliminary version of the system. There are a couple of issues to address:

1) The 17 participants who evaluate the system come from different disciplines. It is not clear if children are involved or only their parents, or if S/W professionals are involved. Please clerify and check for differences among the different groups of participants.

2) The pictorial part seems the most interesting one. In my view the authors should propose a way to navigate and manage pictorial content in a more automated way (e.g. AI/DL/ML driven) from this stage if the paper is to be published in Applied Sciences.

Author Response

We are grateful for the reviewer's comments and suggestions for changes that will improve the submitted article. All changes in the manuscript have been highlighted in blue.

This paper presents an IT approach for managing children with ASD as it concerns their QoL and the continuous communication with their environment, informal carers and physicians.

The topic is interesting, but it seems that this paper shows a preliminary version of the system.

There are a couple of issues to address:

1) The 17 participants who evaluate the system come from different disciplines. It is not clear if children are involved or only their parents, or if S/W professionals are involved. Please clarify and check for differences among the different groups of participants.

Authors> We have tried to clarify, through the article, that the assessment did not involve ASD children but their relatives or caregivers (lines 105, 276-277, 283-284, 306-314) Indeed, it has also been focused in the limitations section (lines 421-423), in the discussion (lines 491-494), and in the conclusions (lines 516-518).

Regarding the differences between the three groups of participants, a new table (Table 4) has been included showing the answers given to the items and dimensions of the evaluation questionnaires, as well as a new paragraph commenting on the results obtained. 

2) The pictorial part seems the most interesting one. In my view the authors should propose a way to navigate and manage pictorial content in a more automated way (e.g. AI/DL/ML driven) from this stage if the paper is to be published in Applied Sciences.

Authors> We appreciate this interesting suggestion and have included a sentence (lines 491-495) in the discussion section which reflects our intention to consider it in the future as it is impossible for us to address it at this time: it is beyond the limits and scope of this research work.

Reviewer 3 Report

Dear Molina et al.,

The manuscript “PlanTEA: Supporting planning and anticipation for children with ASD attending medical appointments” (applsci-1704565) by Molina et al. incorporates entertainment mechanisms during waiting times, positive reinforcement methods (rewards) and will facilitate, thanks to the inclu-sion of an augmentative and alternative communication booklet, the communication with medical staff, when necessary. The topic is interesting, but I think this article should reconsider after proper changes in major revision for publication in Applied Sciences. Some of my specific comments are below:

  1. In the abstract section (line 16-33), the authors should add quantitative results rather than only qualitative results.
  2. Describe the novelty of the article made by the author? From the results of my evaluation, it seems that many similar published works adequately explain what you have raised in the current manuscript related to software application for supporting children with autism spectrum disorder. If there are something others really new in this manuscript, please highlight it more clearly in the introduction section (line 37-102).
  3. Since this manuscript related to autism spectrum disorder, I would encourage and advise the authors to adopt some of the specific additional references related to autism spectrum disorder in the introduction section (line 37-102) as follow:
    • Physiological Effect of Deep Pressure in Reducing Anxiety of Children with ASD during Traveling: A Public Transportation Setting. Bioengineering 2022, 9, 157. https://doi.org/10.3390/bioengineering9040157
    • Effect of Short-Term Deep-Pressure Portable Seat on Behavioral and Biological Stress in Children with Autism Spectrum Disorders: A Pilot Study. Bioengineering 2022, 9, 48. https://doi.org/10.3390/bioengineering9020048
    • The Subjective Comfort Test of Autism Hug Machine Portable Seat. J. Intellect. Disabil. - Diagnosis Treat. 2021, 9, 182–188. https://doi.org/10.6000/2292-2598.2021.09.02.4
  1. The participant involved in the software development is very small that would bring to bias target and misinterpretation. May the authors explain and support its limitation regarding participant involved? Or orthers similar project related to support the use of small participant involved.
  2. Limitation in the present manuscript (line 365-400) are recommended to writted as paragraph rather than poin-by-point.
  3. The conclusion (line 466-495) of the present manuscript is too long and not solid. Further elaboration is needed.
  4. In the whole of the manuscript, the authors sometimes made a paragraph only consisting of one or two sentences that made the explanation not clearly understood. The authors need to extend their explanation to become a more comprehensive paragraph. In one paragraph, it is recommended to consist of at least 3 sentences with 1 sentence as the main sentence and the other sentences as supporting sentences.
  5. Please make sure the authors have used the Applied Science, MDPI format correctly. The authors can download published manuscripts by Applied Science, MDPI, and compare them with the present author's manuscript to ensure typesetting is appropriate.

I am pleased to have been able to review the author's present manuscript. Hopefully, the author can revise the current manuscript as well as possible so that it becomes even better. Good luck for the author's work and effort.

Best regards,

The Reviewer

Author Response

We are grateful for the reviewer's comments and suggestions for changes that will improve the submitted article. All changes in the manuscript have been highlighted in blue.

The manuscript “PlanTEA: Supporting planning and anticipation for children with ASD attending medical appointments” (applsci-1704565) by Molina et al. incorporates entertainment mechanisms during waiting times, positive reinforcement methods (rewards) and will facilitate, thanks to the inclu-sion of an augmentative and alternative communication booklet, the communication with medical staff, when necessary. The topic is interesting, but I think this article should reconsider after proper changes in major revision for publication in Applied Sciences.

Some of my specific comments are below:

1. In the abstract section (line 16-33), the authors should add quantitative results rather than only qualitative results.

Authors> Following the reviewer's recommendation, we have included some quantitative results in the abstract.

2. Describe the novelty of the article made by the author? From the results of my evaluation, it seems that many similar published works adequately explain what you have raised in the current manuscript related to software application for supporting children with autism spectrum disorder. If there are something others really new in this manuscript, please highlight it more clearly in the introduction section (line 37-102).

Authors> A new paragraph has been included in the introduction which highlights the novelty of the proposal described in this article.

3. Since this manuscript related to autism spectrum disorder, I would encourage and advise the authors to adopt some of the specific additional references related to autism spectrum disorder in the introduction section (line 37-102) as follow:

    • Physiological Effect of Deep Pressure in Reducing Anxiety of Children with ASD during Traveling: A Public Transportation Setting. Bioengineering 2022, 9, 157. https://doi.org/10.3390/bioengineering9040157
    • Effect of Short-Term Deep-Pressure Portable Seat on Behavioral and Biological Stress in Children with Autism Spectrum Disorders: A Pilot Study. Bioengineering 2022, 9, 48. https://doi.org/10.3390/bioengineering9020048
    • The Subjective Comfort Test of Autism Hug Machine Portable Seat. J. Intellect. Disabil. - Diagnosis Treat. 2021, 9, 182–188. https://doi.org/10.6000/2292-2598.2021.09.02.4

Authors> We thank the reviewer for the proposed bibliographical references, which have been incorporated and referenced in the introduction section, and have served to better justify the need for support in everyday situations for people with ASD.

4. The participant involved in the software development is very small that would bring to bias target and misinterpretation. May the authors explain and support its limitation regarding participant involved? Or orthers similar project related to support the use of small participant involved.

Authors> The main scope of the presented article is the description of the PlanTEA tool, its design process and a preliminary evaluation. As mentioned in the limitations (lines 420-428), discussion (lines 496-499) and conclusion sections (lines 516-522), evaluation with end-users (people with ASD) is a work, in which we are currently working on. We are planning and preparing for a larger scale evaluation, which will apply the methodology and evaluation protocol described in:

Valencia, K.; Rusu, C.; Botella, F. A Preliminary Methodology to Evaluate the User Experience for People with Autism Spectrum Disorder. In International Conference on Human-Computer Interaction; Springer, 2021; pp 538–547.

The description of this evaluation and the results obtained are beyond the limits and scope of this article and are the subject of a subsequent publication on which work is currently being carried out.

In any case, in the limitations section of the article (4.4) we have pointed out that the size of the sample participating in the evaluation may lead to biases in the interpretation of the results, as noted by the reviewer.

In the last paragraph of the discussion section, it is pointed out that the evaluation of systems aimed at the group of users with ASD is a recurrent and common problem, supporting this statement with relevant bibliographical references.

5. Limitation in the present manuscript (line 365-400) are recommended to writted as paragraph rather than poin-by-point.

Authors> The limitations section has been rewritten, removing the bulleted enumeration and changing the wording to paragraphs. 

6. The conclusion (line 466-495) of the present manuscript is too long and not solid. Further elaboration is needed.

Authors> The conclusions section has been rewritten.

7. In the whole of the manuscript, the authors sometimes made a paragraph only consisting of one or two sentences that made the explanation not clearly understood. The authors need to extend their explanation to become a more comprehensive paragraph. In one paragraph, it is recommended to consist of at least 3 sentences with 1 sentence as the main sentence and the other sentences as supporting sentences.

Authors> We have revised the content, structure and length of the paragraphs of the article to take into account the reviewer's comments and suggestions for improvement. The shorter paragraphs have been appropriately connected with the rest of the paragraphs to facilitate the reading of the content.

8. Please make sure the authors have used the Applied Science, MDPI format correctly. The authors can download published manuscripts by Applied Science, MDPI, and compare them with the present author's manuscript to ensure typesetting is appropriate.

Authors> The journal and editorial formatting has been rechecked to ensure that it complied correctly.

I am pleased to have been able to review the author's present manuscript. Hopefully, the author can revise the current manuscript as well as possible so that it becomes even better. Good luck for the author's work and effort.

Authors> Thank you very much for your comments and suggestions, which will undoubtedly improve the final version of the article.

Round 2

Reviewer 2 Report

Overal the manuscript is much improved. The authors show that they are aware of the limitations of the study. On the other hand Ithink that Tables 2,3 and especially yTable 4 have helped the authors show the part of innovation that corresponds to this study. E.g. differences in the evaluation metrics of ICT people vs. parents who live with the problem. Can they bridge this gap and provide a better system? This is partly addressed in the discussion. As it concerns the reference to AI/DL the authors correctly refer to a needed ICT approach but they will need to add some references e.g. related with picture rating tasks and visual analytics.

Author Response

We are grateful for the reviewer's comments and suggestions for changes that will improve the submitted article. All changes made to the article, as well as content that allows for a reply to the reviewer's comments, have been highlighted in blue.

Overall the manuscript is much improved. The authors show that they are aware of the limitations of the study. On the other hand I think that Tables 2,3 and especially y Table 4 have helped the authors show the part of innovation that corresponds to this study. E.g. differences in the evaluation metrics of ICT people vs. parents who live with the problem. Can they bridge this gap and provide a better system? This is partly addressed in the discussion.

Authors> In order to solve usability problems, and a better acceptance by the final users of the application (users with ASD and their families), specific design guidelines for autistic users (AutismGuide) have been applied (lines 96-100; 117-120; 222-223):

  • Aguiar, Y. P. C.; Galy, E.; Godde, A.; Trémaud, M.; Tardif, C. AutismGuide: A Usability Guidelines to Design Software Solutions for Users with Autism Spectrum Disorder. Behaviour and Information Technology. 2020. https://doi.org/10.1080/0144929X.2020.1856927.

Currently we are working on the redesign of PlanTEA, to take into account the recommendations, extensions and proposed changes obtained from the evaluation described in this paper (lines 487-490). In the redesign process, the level of compliance with the AutismGuide guidelines will be reviewed and additional checklists and usability recommendations for ASD users will be applied:

  • Valencia, K.; Rusu, C.; Botella, F. User Experience Factors for People with Autism Spectrum Disorder. Appl. Sci. 2021, 11 (21). https://doi.org/10.3390/app112110469.

Furthermore, as a future work that is already being carried out, an evaluation process of the system with end users will be conducted, in which a specific evaluation methodology for applications targeted at ASD users will be followed (lines 496-499; 518-522):

  • Valencia, K.; Rusu, C.; Botella, F. A Preliminary Methodology to Evaluate the User Experience for People with Autism Spectrum Disorder. In International Conference on Human-Computer Interaction; Springer, 2021; pp 538–547.

Therefore, both in the development and evaluation of PlanTEA, recommendations and design guidelines adapted to the target group of users are applied, aiming to reduce the gap mentioned by the reviewer.

As it concerns the reference to AI/DL the authors correctly refer to a needed ICT approach but they will need to add some references e.g. related with picture rating tasks and visual analytics.

Authors> Two current bibliographical references on visual analytical techniques have been included.

Reviewer 3 Report

Dear Molina et al.,

After carefully reading the author's revised manuscript entitled "PlanTEA: Supporting planning and anticipation for children with ASD attending medical appointments" (applsci-1704565) by Molina et al., The authors have made significant improvements in the revised manuscript. Also, all of the issues in my review report have been addressed precisely.

With my pleasure, I recommend the manuscript should be accepted for publication on Applied Sciences.

Best regards,

The Reviewer

Author Response

Dear Molina et al.,

After carefully reading the author's revised manuscript entitled "PlanTEA: Supporting planning and anticipation for children with ASD attending medical appointments" (applsci-1704565) by Molina et al., The authors have made significant improvements in the revised manuscript. Also, all of the issues in my review report have been addressed precisely.

With my pleasure, I recommend the manuscript should be accepted for publication on Applied Sciences.

Best regards,

The Reviewer

Authors> We are grateful for the reviewer's comments and suggestions for changes that have helped to improve the article.
